# Probabilistic Dual-Space Fusion for Real-Time Human-Robot Interaction

**DOI:** 10.3390/biomimetics8060497

**Published:** 2023-10-19

**Authors:** Yihui Li, Jiajun Wu, Xiaohan Chen, Yisheng Guan

**Affiliations:** 1Biomimetic and Intelligent Robotics Lab (BIRL), Guangdong University of Technology, Guangzhou 510006, China; liyihui.ing@gmail.com (Y.L.); 2112101012@mail2.gdut.edu.cn (J.W.); 2112001215@mail2.gdut.edu.cn (X.C.); 2State Key Laboratory of Precision Electronic Manufacturing, Guangzhou 510006, China

**Keywords:** dual-space fusion, physical human-robot interaction, real-time HRI, probabilistic learning, robot learning, learning from demonstrations, imitation learning

## Abstract

For robots in human environments, learning complex and demanding interaction skills from humans and responding quickly to human motions are highly desirable. A common challenge for interaction tasks is that the robot has to satisfy both the task space and the joint space constraints on its motion trajectories in real time. Few studies have addressed the issue of hyperspace constraints in human-robot interaction, whereas researchers have investigated it in robot imitation learning. In this work, we propose a method of dual-space feature fusion to enhance the accuracy of the inferred trajectories in both task space and joint space; then, we introduce a linear mapping operator (LMO) to map the inferred task space trajectory to a joint space trajectory. Finally, we combine the dual-space fusion, LMO, and phase estimation into a unified probabilistic framework. We evaluate our dual-space feature fusion capability and real-time performance in the task of a robot following a human-handheld object and a ball-hitting experiment. Our inference accuracy in both task space and joint space is superior to standard Interaction Primitives (IP) which only use single-space inference (by more than 33%); the inference accuracy of the second order LMO is comparable to the kinematic-based mapping method, and the computation time of our unified inference framework is reduced by 54.87% relative to the comparison method.

## 1. Introduction

As robotics advance, human-robot integration and human-robot interaction (HRI) have emerged as new research directions. To enable robots to seamlessly integrate into human daily life, it is essential to study HRI technology, especially physical human-robot interaction (pHRI) [1]. One of the main research challenges is how to make robots learn the skills to interact with humans more effectively. A popular research topic is to make robots learn collaborative skills through an imitative learning approach. This approach allows robots to generalize a skill or task by imitating it, thus avoiding complex expert programming [2,3]. Common imitation learning methods include the Hidden Markov Model (HMM) [4], the Gaussian Mixture Model (GMM) [5], the Dynamical Movement Primitives (DMP) [6], the Probabilistic Movement Primitives (ProMP) [7], and the Kernelized Movement Primitives (KMP) [8], etc.

Learning skills from demonstration samples is similar to a martial arts practitioner imitating a master’s movements, and a proficient student can acquire various features from the master’s demonstrations. Likewise, researchers expect robots to learn more features from the demonstration samples. To address robot skill learning, researchers have proposed the concept of dual-space feature learning [9], where dual space usually refers to the task space and the joint space of the robot. Task space is a description of robot actions in Cartesian coordinates, and joint space is a description of robot actions in terms of joint angles. To perform complex and demanding tasks, the robot needs to learn different characteristics of skillful actions from both task space and joint space perspectives [10,11,12]. For example, when a robot imitates a human to generate the motion of writing brush characters, it should not only match the end-effector trajectory to the writing specification, but also adjust the joint posture to resemble the human’s. Another example is when the robot operates in a dynamic environment where obstacles or targets may change their positions or orientations. By integrating the learned knowledge of both task space and joint space from demonstrations, the robot can adapt its motion to the changing situation and avoid collisions or reach the desired goal. Therefore, the learning algorithm should extract and fuse both end-effector and joint posture features from the demonstrated action.

These requirements are not only in the field of skill imitation learning but also in the field of HRI where it is desirable for the robot to learn more action features. In our previous work [13], we proposed a method for simultaneous generalization and probabilistic fusion of dual-space trajectories, which enables the robot to simultaneously satisfy the dual-space constraints in an HRI task. We demonstrated the feasibility of the method through experiments in which UR5 followed a human-held object. This case required that the robot end-effector followed the target object in the task space, whereas the joint pose needed to move in a specific pattern so as to bypass the box next to it. However, the approach relies on the robot model and requires high computation of the robot Jacobian matrix for each time step. Therefore, in this paper, we extend our previous work and propose a method that does not rely on a robot model, making it more general. In addition, the learning of multiple features is essentially the establishment and inference of a multidimensional model, and with an increase in model dimensionality, the computational performance of the algorithm will be affected to some extent. Therefore, this paper carries out further research and enhancement on the real-time performance of the HRI method. The overview of the proposed framework is shown in Figure 1. Furthermore, our contributions are mainly threefold:A physical human-robot interaction method with dual-space (task space and joint space) features fusion is proposed, which improves the accuracy of the inferred trajectories in both spaces.We also propose a data-driven linear mapping operator for mapping from task space to joint space, which does not rely on robot models and time-consuming Jacobian computation, and is suitable for HRI scenarios especially those involving high-dimensional modeling.We present a unified probabilistic framework for integrating dual-space trajectories fusion, linear mapping operator and phase estimation, which scales particularly well with high dimensionality of the task and reduces the inference computation.

The rest of the paper is organized as follows. We begin by reviewing the previous related work on motion generation methods and spaces fusion in the field of HRI in Section 2. In Section 3, Section 4 and Section 5, we outline the methodology of our framework including dual-space fusion, linear mapping operator, phase estimation, and the unified framework. We review and discuss our experimental setups and results in Section 6. Finally, we conclude our work in Section 7.

## 2. Related Work

The primary problem in pHRI is generating robot motion by observing human motion. Amor et al. [14] proposed Interaction Primitives (IP) based on DMP for modeling the relationship between human and robot. IP is to fit the human trajectory and the robot trajectory separately using the DMP method, then combine the DMP parameters of the two trajectories, and finally observe the human trajectory and infer and predict the robot trajectory. IP has been applied to multiple scenarios [15,16]. However, this method only models the spatial uncertainty in the human-robot interaction process and does not model the temporal uncertainty. Therefore, it is difficult to avoid the problem of time synchronization in its interaction process. At the same time, methods based on the DMP framework need to select the type and number of basis functions. As the dimension increases, the number of basis functions also increases, and the complexity of the inference calculation process also increases. Huang et al. proposed KMP [8] to take the human motion sequence as the input state and directly predict the robot trajectory. Similar to the KMP method, Silvério et al. [17] performed HRI through Gaussian Process (GP), which also avoided the problem of time synchronization, but this method did not consider the covariance and model generalization of multi-dimensional trajectories. Vogt et al. [18,19] used GMM and HMM to model the interaction between humans and humans, and proposed the Interactive Meshes (IMs) method, which mapped the spatiotemporal relationship of teaching motion to the interaction between humans and robots. Wang et al. [20] utilized a Gaussian Process (GP) dynamics model to infer human intention based on human behavior and applied it to ping-pong robots. Ikemoto et al. [21] applied GMM to the scenario of physical contact between humanoid robots and humans. Although these works use many mature theories to solve the problems of temporal and spatial uncertainty involved in robot learning, they almost all consider a single spatial feature, either joint space or task space, without involving the fusion of both. Therefore, when the task involves multiple spatial requirements, such as considering joint avoidance during robot trajectory tracking [22,23], these methods are not competent.

Dual-space trajectories fusion refers to learning the movement features of different spaces in demonstrations by fusing the movements of a robot’s task space and joint space, and the purpose is to complete the robot operation tasks under the joint constraints of both spaces. In the field of HRI, most current works only focus on the single space of robotics, e.g., Chen et al. proposed an HRI framework in joint space without constraining the task space [24]. Li et al. [25] proposed a topology-based motion generalization method for the general motion generation problem, which maintains the spatial relationship between different parts of a robot or between different agents with a topology representation; the configuration prior (joint space) is preserved in the generated motion, and the target position (task space) of the end-effector also can also be realized by applying certain constraint. However, the process of generating these two space features is separative and there is no fusion essentially. Calinon et al. [22] used the relative distance trajectory between the robot and the object as the task space trajectory and modeled it. Specifically, the task space trajectory and joint space trajectory of the robot in the demonstrations are modeled with GMM separately. Then Gaussian Mixture Regression (GMR) is leveraged to obtain the probability distribution of the two spatial trajectories, and finally Gaussian features are used to fuse the two trajectories.Schneider et al. [26] adopted a similar way to perform dual-space feature learning of robot trajectories, but replaced the modeling method of GMM and GMR with Heteroscedastic Gaussian Processes (HGP) [27,28]. However, Calinon and Schneider’s work did not perform joint modeling of the two spatial trajectories, and could not achieve synchronous generalization of the two spaces. Its essence is still to generalize the joint space trajectory. In addition, their method cannot guarantee that the generalization directions of different spatial trajectories are consistent, which means that the generalization result for one space may not be beneficial to another space. To this end, Huang et al. [29] presented a work for solving the problem of dual-space synchronous generalization. They considered the generalization of task space while using null space to generalize robot joint space trajectory, but they did not consider the problem of interaction between humans and robots, and could not be used for real-time interaction scenarios.

Regarding the real-time performance of those methods, KMP can model the trajectory uncertainty outside the range of the demonstrations, and the kernel function processing method in the KMP inference process is similar to the Heteroscedastic Gaussian Process, its computational complexity is $O(N^{3})$. It cannot adapt well in the case of long trajectory sequences and has poor real-time performance. Campbell et al. proposed Bayesian Interaction Primitives (BIP) [30] and the improved BIP [31] inspired by the related method of robot pose estimation and update in Simultaneous Localization and Mapping (SLAM). Specifically, BIP is essentially based on the work of Amor et al. [14] and Maeda et al. [32], and applies Extended Kalman Filter (EKF) to the phase estimation of Interaction Primitives, making the phase estimation of Interaction Primitives and Bayesian inference [33] process synchronized, and improving computational efficiency. The time complexity of the time alignment method in the original IP framework is $O(N^{2})$, BIP removes the extra time alignment process, reducing the interaction inference time by 66%.

As shown in Table 1, most of the existing research on pHRI or skill learning focuses on either task space or joint space, but not both. Moreover, few methods can achieve dual-space fusion and generalization, as well as meet the real-time interaction requirements. However, the existing methods have advanced in modeling spatial and temporal uncertainties, phase estimation, and other related theories, which can provide guidance for our further exploration.

## 3. Preliminaries: Probabilistic Fusion in Task Space and Joint Space

### 3.1. Simultaneous Generalization of Dual Space

The goal of the simultaneous generalization of dual space is to enable the robots (the controlled agent in pHRI) to learn the features of the motion trajectories in dual space. This enables the robot to perform complex tasks, such as tracking a target object while avoiding obstacles. To achieve the simultaneous generalization of dual space, two components are needed: the learning phase and the inference phase. The former aims to learn the features of trajectories in dual space. The latter aims to generate the robot’s motion trajectories in dual space based on the observed human motion trajectories.

In the learning phase, we propose a probabilistic fusion model of task space and joint space, inspired by IP [14]. This model encodes the human and robot motion trajectories with basis functions and models their spatial and temporal uncertainty with probabilistic weights. We define the joint trajectory of human and robot $f_{1:T}={\left[\xi_{1:T}^{H},\xi_{1:T}^{R}\right]}^{⊤}$ as a sequence of sensor observations of length *T*. Here, $\xi_{1:T}^{H}$ is a sequence of human movement trajectories with $D_{H}$ degrees of freedom. For a time step *t*, we denote $\xi_{t}^{H}=\left[{\left.y_{1}^{H}\right|}_{t},{\left.y_{2}^{H}\right|}_{t},\ldots ,{\left.y_{D_{H}}^{H}\right|}_{t}\right]$ as the human trajectory at that time step, or simply $\xi_{t}^{H}=\left[y_{1}^{H},y_{2}^{H},\ldots ,y_{D_{H}}^{H}\right]$ for brevity. $\xi_{t}^{R}=\left[\xi_{t}^{R_{t}},\xi_{t}^{R_{j}}\right]$ is a sequence of robot movement trajectories with the number of degrees of freedom $D_{R}=D_{R_{t}}+D_{R_{j}}$ at time step *t*. $\xi_{t}^{R_{t}}$ is the trajectory point of the robot in task space at time step *t*, $\xi_{t}^{R_{j}}$ is the trajectory point of the robot in joint space, and $D_{R_{t}}$ and $D_{R_{j}}$ are the numbers of degrees of freedom of the two space trajectories.

The sequence for each degree of freedom is linearly fitted using *B* basis functions containing Gaussian white noise. This fitting method is derived from DMP: (1)$$y_{d}=\Phi^{⊤}\left(t\right)w_{d}+\varepsilon_{\xi }$$
where $\Phi^{⊤}\left(t\right)$ is the vector composed of *B* Gaussian basis functions, $w_{d}$ is the weight corresponding to the basis function, and $\varepsilon_{\xi }$ is the corresponding Gaussian white noise.

The joint weight matrix $w={\left[w^{H},w^{R_{t}},w^{R_{j}}\right]}^{⊤}$ is obtained by fitting using Equation (Equation 1). It is assumed that the weights learned from each demo follow a Gaussian distribution, i.e., $w∼N(\mu_{w},\Sigma_{w})$, and the Gaussian parameters $\mu_{w}\in R^{DB\times \;1}$ and $\Sigma_{w}\in R^{DB\times DB}$ can be obtained from multiple demos by Maximum Likelihood Estimation (MLE), where *D* is the sum of the dimensions of the human movement trajectories and the robot dual space movement trajectories, and *B* is the number of Gaussian basis functions. Furthermore, it is important to note that $\Sigma_{w}$ captures the correlation among human motion trajectories, robot motion trajectories in task space, and robot motion trajectories in joint space.

In the inference process of HRI, we use the weights learned from the learning phase as prior information. We observe the human motion trajectory $\xi_{1:t^{*}}^{H}$ from the beginning of the interaction to time $t^{*}$ and update the entire weight matrix using Bayesian recursive inference.
(2)$$p\left(w|\xi^{*}\right)\propto \;p\left(\xi^{*}|w\right)p\left(w\right)$$

Then, we use the $w^{R_{t}}$ and $w^{R_{j}}$ components of the inferred weight matrix w to generate the robot motion trajectories in dual space according to Equation (Equation 1). So for an observation time step $t^{*}$, we denote the generated task space trajectory as $\xi_{t^{*}:T}^{R_{t}}$, and the joint space trajectory as $\xi_{t^{*}:T}^{R_{j}}$, where $R_{t}$ and $R_{j}$ represent robot task space and robot joint space, respectively.

### 3.2. Kinematics-Based Linear Mapping

We have obtained two control trajectories for the robot in both task space and joint space until now, but, in fact, we only need one controller to control the motion of the robot. Therefore, a mapping from task space to joint space for the trajectory is required, and then fusing the two joint space trajectories into one in the form of probability distributions. The most common mapping method in robotics is inverse kinematics with a kind of Jacobian matrix. So, based on the kinematic relationship of the robot, the task space trajectory sequence $\xi_{t^{*}:T}^{R_{t}}$ is linearly and iteratively mapped into the joint space.

Assume that the trajectory point $\xi_{t}$ at each time step *t* ($t\in \left[t^{*},T\right]$), follows a Gaussian distribution, i.e., $\xi_{t}∼N\left(\mu_{t},\Sigma_{t}\right)$, and obviously the parameters $\mu_{t}$ and $\Sigma_{t}$ can be obtained by performing a linear transformation (Equation (Equation 1)) on the updated weight distribution $N(\mu_{w}^{*},\Sigma_{w}^{*})$, as well as the original and mapped joint space trajectory point $\xi_{t}^{R_{j}}∼N\left(\mu_{t}^{R_{j}},\Sigma_{t}^{R_{j}}\right)$ and $\xi_{t}^{{R_{j}}^{*}}∼N\left(\mu_{t}^{{R_{j}}^{*}},\Sigma_{t}^{{R_{j}}^{*}}\right)$. The new trajectory obtained from the kinematics-based mapping is denoted as $\xi_{t^{*}:T}^{R_{j}^{*}}$, and the update formula of a point of the new trajectory at time step *t* is:(3)$$\xi_{t+1}^{R_{j}^{*}}=\xi_{t}^{R_{j}^{*}}+J^{†}\left(\xi_{t}^{R_{j}^{*}}\right){\dot{\xi }}_{t}^{R_{t}}$$
(4)$$\Sigma_{t+1}^{R_{j}^{*}}=\Sigma_{t}^{R_{j}^{*}}\;+\;J^{†}\left(\xi_{t}^{R_{j}^{*}}\right)\;{\dot{\Sigma }}_{t}^{R_{t}}\;J^{†}{\left(\xi_{t}^{R_{j}^{*}}\right)}^{⊤}$$
where $J^{†}\left(\xi_{t}^{R_{j}^{*}}\right)$ is the pseudo-inverse of the robot Jacobian matrix $J\left(\xi_{t}^{R_{j}^{*}}\right)$ at time step *t*, and ${\dot{\xi }}_{t}^{R_{t}}$ is the derivative of $\xi_{t}^{R_{t}}$. Since $\xi_{t}^{R_{j}}$ follows a Gaussian distribution, ${\dot{\xi }}_{t}^{R_{t}}$ does so as well, i.e., ${\dot{\xi }}_{t}^{R_{t}}∼N\left({\dot{\mu }}_{t}^{R_{t}},{\dot{\Sigma }}_{t}^{R_{t}}\right)$. In addition, the observed joint angle $\xi_{t^{*}}^{R_{j}}$ at the time step of $t^{*}$ is taken as the initial value of the iteration, that is $\xi_{t^{*}}^{R_{j}^{*}}=\xi_{t^{*}}^{R_{j}}$.

### 3.3. Probabilistic Fusion

From the above iterative mapping method, two kinds of movement trajectories in robot joint space can be obtained, one of which is the robot joint space trajectory sequence $\xi_{t^{*}:T}^{R_{j}}$ obtained by direct extrapolation from the human action sequence $\xi_{1:t^{*}}^{H}$, and the other is the joint space sequence $\xi_{t^{*}:T}^{R_{j}^{*}}$ mapped from the robot task space trajectory $\xi_{t^{*}:T}^{R_{t}}$. At time step *t*, there are $\xi_{t}^{R_{j}}∼\;N\left(\mu_{t}^{R_{j}},\Sigma_{t}^{R_{j}}\right)$ and $\xi_{t}^{R_{j}^{*}}∼\;N\left(\mu_{t}^{R_{j}^{*}},\Sigma_{t}^{R_{j}^{*}}\right)$.

$\xi_{t^{*}:T}^{R_{j}}$ and $\xi_{t^{*}:T}^{R_{j}^{*}}$ can be regarded as two probability estimates obtained from the same space (joint space) for the same observation object (human motion sequence $\xi_{1:t^{*}}^{H}$). Therefore, the trajectory sequence $\xi_{t^{*}:T}^{R_{j}}$ and $\xi_{t^{*}:T}^{R_{j}^{*}}$ are two estimation results of the same estimation object (joint space control trajectory, denoted by ${\hat{\xi }}_{j}^{R}$), whereas $\xi_{t^{*}:T}^{R_{j}^{*}}$ contains the information of task space feature generalization, $\xi_{t^{*}:T}^{R_{j}}$ contains the information of joint space feature generalization, and by fusing the two feature information at each time step, we can obtain the control trajectory that contains the features of both spaces.

The two estimates are considered to be independent of each other, and the distributions of the two trajectory estimates are fused at each time step *t* with Bayesian estimation: (5)$$p\left({\hat{\xi }}_{t}^{R_{j}}|\xi_{t}^{R_{j}},\xi_{t}^{R_{j}^{*}}\right)=\eta_{t}p\left({\hat{\xi }}_{t}^{R_{j}}|\xi_{t}^{R_{j}}\right)p\left({\hat{\xi }}_{t}^{R_{j}}|\xi_{t}^{R_{j}^{*}}\right)$$
(6)$$\eta_{t}=\frac{p\left(\xi_{t}^{R_{j}}\right)p\left(\xi_{t}^{R_{j}^{*}}\right)}{p\left(\xi_{t}^{R_{j}},\xi_{t}^{R_{j}^{*}}\right)p\left({\hat{\xi }\;}_{t}^{R_{j}}\right)}$$

At time step *t*, the control trajectory of the robot in the joint space follows a Gaussian distribution $N\left({\hat{\mu }}_{t}^{R_{j}},\;{\hat{\Sigma }}_{t}^{R_{j}}\right)$, with: (7)$${\hat{\mu }}_{t}^{R_{j}}=\frac{\mu_{t}^{R_{j}^{*}}\Sigma_{t}^{R_{j}}+\mu_{t}^{R_{j}}\Sigma_{t}^{R_{j}^{*}}}{\Sigma_{t}^{R_{j}}+\Sigma_{t}^{R_{j}^{*}}}$$

Finally, we obtain the robot control trajectory ${\hat{\xi }}_{t}^{R_{j}}={\hat{\mu }}_{t}^{R_{j}}$, which includes the fusion features of both task space and joint space.

## 4. Kinematics-Free Linear Mapping Method

The mapping relationship between task space and joint space in the last subsection is derived from the robot kinematic model and relies on the Jacobian matrix computed from the robot model. We would like to make the application of the dual-space probabilistic fusion method in robot interaction skill learning more flexible and extend it to more non-kinematic scenarios. For example, we can map and fuse only a part of the interaction data in task space or joint space, or we can apply the method to other spatial fusion tasks that do not involve task space or joint space. However, the computation of the Jacobian matrix in inverse kinematics is time-consuming, so it is not suitable for HRI tasks with high real-time requirements. In this part, we will introduce a linear mapping operator (LMO) for this purpose.

### 4.1. Linear Mapping Operators

We propose a linear mapping approach between two spaces based on Jacobian-like matrices, which are defined as operators that act on data (displacement) vectors. We generalize the notion of dual-space mapping beyond the conventional task and joint spaces and introduce two abstract function spaces, denoted as space *A* and space *B*. Assume that, given demonstration samples, we can learn a time-varying LMO denoted as $L_{t}$ for each time step *t* of HRI, such that it maps vectors from space *A* to space *B*, i.e., A→B.

A possible way to model and estimate the LMO ($L_{t}$) is to treat it as a dynamical system that varies over time. However, this approach may not capture the exact nature of the operator, because it may involve different mapping spaces and nonlinear or discrete transformations in the complex case of dynamical parameters. Therefore, we do not assume a specific mathematical model for the operator but rather regard it as a time-varying parameter to be estimated. There have been many studies on the estimation of time-varying system parameters, e.g., Cui et al. [35] used the Kalman Filter (KF) method to estimate the system state, Kim et al. [36] used the Extension Kalman Filter (EKF) method to estimate the nonlinear system, and Campbell et al. [30] applied the EKF to the trajectory weight inference of Interaction Primitives in HRI. We propose a framework using the Kalman estimation to jointly express the linear mapping operator and the robot interaction trajectory inference.

### 4.2. Modeling of Linear Mapping Operators

Let L, $\xi^{A}$, and $\xi^{B}$ be the time-dependent state sequences corresponding to $L_{t}$, $\xi_{t}^{A}$, and $\xi_{t}^{B}$, respectively. For an HRI process with a time length of T, the trajectories corresponding to $L_{t}$, $\xi_{t}^{A}$, and $\xi_{t}^{B}$ can be denoted as $L_{0:T}$, $\xi_{0:T}^{A}$, $\xi_{0:T}^{B}$, respectively. The mathematical relationship between $L_{t}$, $\xi_{t}^{A}$, and $\xi_{t}^{B}$ is expressed as follows: (8)$$\xi_{t}^{A}=L_{t}\xi_{t}^{B},\;t\in \left[0,T\right]$$
which denotes the trajectory point of space B mapped into space A at time step *t*, i.e., $\xi_{t}^{B}\to \xi_{t}^{A}$, where $\xi_{t}^{A}\in R^{m\times 1}$, $\xi_{t}^{B}\in R^{n\times 1}$, $L_{t}\in R^{m\times n}$, where *m* and *n* is the number of dimensions of space A and space B, respectively, (a more specified definition for an HRI system is introduced in Section 3.1). The above equation can obtain: (9)$$L_{t}=\left(\xi_{t}^{A}{\xi_{t}^{A}}^{⊤}\right){\left(\xi_{t}^{B}{\xi_{t}^{A}}^{⊤}\right)}^{†}$$

Although $L_{t}$ is the linear mapping relationship between two spaces, it is assumed to be a nonlinear dynamic system itself, and needs to be linearly approximated by Taylor expansion or Taylor series, and its expansion order affects the accuracy of model establishment. Theoretically, the higher the expansion order, the higher the mapping accuracy of the model, but it will also bring about a decrease in inference efficiency. Let *o* be the order of linear mapping operator expansion, then $L_{t}$ can be discretized and expressed as: (10)$$L_{t}\approx L_{t-1}+\frac{{\partial L}_{t-1}}{\partial t}\Delta t+\frac{\partial \left(\frac{{\partial L}_{t-1}}{\partial t}\right)}{\partial t}\frac{\Delta t^{2}}{2}+\cdots +\frac{\partial L_{t-1}^{o-1}}{\partial t}\frac{\Delta t^{o}}{o!},$$
where $\frac{{\partial L}_{t-1}}{\partial t}$ and $\frac{\partial \left(\frac{{\partial L}_{t-1}}{\partial t}\right)}{\partial t}$ is the first and sencond partial derivative of $L_{t-1}$ with respect to *t*, respectively, marked as ${L_{t}}^{{}^{\prime }}$ (or ${L_{t}}^{1}$) and ${L_{t}}^{{}^{\prime \prime }}$ (or ${L_{t}}^{2}$). At time step *t*, we assume that the coefficients of each term in the series follows a Gaussian distribution, i.e., $L_{ij(t)}∼N\left(\mu_{L_{ij(t)}},\sigma_{L_{ij(t)}}^{2}\right)$. Therefore, we can estimate the linear mapping operator by estimating the parameters of its finite expansion orders.

In practical applications, we need to learn from the demonstrations to obtain the parameters of the LMO. By observing the demonstration sample data at a certain observation frequency (120Hz in our demonstrations), we can obtain the human and robot motion data sequences in HRI. The LMO at each interaction time step in the demonstration sample is obtained by Equation (Equation 8). The initial value of the model is the mean of the initial LMO in the training sample.

At an observation frequency, the LMO of each sample is discretized at each time step *t*. For example, for the first two orders, we have $L_{t}^{{}^{\prime }}=\frac{L_{t}-L_{t-1}}{\Delta t}$, $L_{t}^{{}^{\prime \prime }}=\frac{L_{t}^{{}^{\prime }}-L_{t-1}^{{}^{\prime }}}{\Delta t}$, where both $L_{t}^{{}^{\prime }}$ and $L_{t}^{{}^{\prime \prime }}$ follows a Gaussian distribution, i.e., $L_{ij(t)}^{{}^{{}^{\prime }}}∼N\left(\mu_{L_{ij(t)}^{{}^{{}^{\prime }}}},\sigma_{L_{ij(t)}^{{}^{{}^{\prime }}}}^{2}\right)$, $\;L_{ij(t)}^{{}^{{}^{\prime \prime }}}∼N\left(\mu_{L_{i+j(t)}^{{}^{{}^{\prime \prime }}}},\sigma_{L_{ij(t)}^{{}^{{}^{\prime \prime }}}}^{2}\right)$.

### 4.3. Inference of Linear Mapping Operators

We adopt a nonlinear KF framework that follows the Bayesian estimation process to infer the LMO and the dual-space weights jointly. If only considering the fusion model and LMO, the state matrix of the model at time *t* is given by: (11)$$s_{t}={\left[\ell_{t}^{0},\ell_{t}^{1},\ldots \ell_{t}^{o},w^{⊤}\right]}^{⊤},\;s_{t}\in R^{D_{w}B+omn},$$
where $\ell_{t}^{i}$ is the vectorized form of $L_{t}^{i}$ (*i* is the order of the expansion, $i\in \left[0,o\right]$), i.e., $\ell_{t}^{i}={\left[\begin{array}{c} L_{11(t)}^{i},\ldots ,L_{mn(t)}^{i} \end{array}\right]}^{⊤}$; w is the joint weight of human and robot motions in two spaces, and $w={\left[w^{H},w^{R_{A}},w^{R_{B}}\right]}^{T}$; $D_{w}$ is the sum of the dimensions of the interaction primitive weights; and *B* is the number of basis functions.

The estimation process aims to compute the probability of the state matrix given an observation sequence $z_{0:t}$, and its conditional probability density function follows a Gaussian distribution $N(s_{t}|\mu_{t},\Sigma_{t})$; we have: (12)$$p\left(s_{t}|z_{0:t}\right)=N\left(s_{t}|\mu_{t},\Sigma_{t}\right),$$
(13)$$\mu_{0}={\left[\mu_{\ell_{0}},\ldots \mu_{\ell_{0}^{o}},\mu_{w}^{⊤}\right]}^{⊤},$$
(14)$$\Sigma_{0}=\left[\begin{array}{cc} \Sigma_{\ell ,\ell } & \Sigma_{\ell ,w}\\ \Sigma_{w,\ell } & \Sigma_{w,w} \end{array}\right],$$
where $\mu_{t}\in R^{D_{w}B+omn}$ and $\Sigma_{t}\in R^{\left(D_{w}B+omn\right)\times \left(D_{w}B+omn\right)}$. The observation object sequence is the HRI trajectory; that is, $z_{0:t}=y_{0:t^{*}}$, $\Sigma_{w,w}=\Sigma_{w}$.

Bayesian filtering methodology consists of two components: the state transition model and the measurement model. Based on the linear approximation assumption of the linear mapping operator in Equation (Equation 10), we can establish the following state transition model for its vectorized form at each time step *t*: (15)$$\mu_{\ell_{t}}=G\mu_{\ell_{t-1}},$$
(16)$$\Sigma_{\ell_{t}}=G\Sigma_{\ell_{t-1}}G^{⊤}+Q_{\ell_{t}},$$
where
$$G=\left[\begin{array}{ccccccc} 1 & \Delta t & \cdots & \frac{\Delta t^{o}}{o!} & 0 & \cdots & 0\\ 0 & 1 & \Delta t & \cdots & \frac{\Delta t^{o}}{o!} & \cdots & 0\\ \vdots & \vdots & \vdots & \ddots & \ddots & \frac{\Delta t^{o}}{o!} & 0\\ 0 & 0 & 0 & \cdots & 0 & 1 & 0 \end{array}\right],$$
$$Q_{t}=\left[\begin{array}{cccc} \sigma_{L,L}^{2} & \sigma_{L,L^{\prime }}^{2} & \cdots & \sigma_{L,L^{o}}^{2}\\ \sigma_{L,L}^{2} & \sigma_{L,L^{\prime }}^{2} & \cdots & \sigma_{L,L^{o}}^{2}\\ \vdots & \vdots & \ddots & \vdots \\ \sigma_{L^{o},L}^{2} & \sigma_{L^{o},L^{{}^{\prime }}}^{2} & \cdots & \sigma_{L^{o},L^{o}}^{2} \end{array}\right].$$

And then we have the measurement model: (17)$$z_{t}=\underset{h(\mu_{t})}{\underset{︸}{\left[\begin{array}{c} \Phi_{\delta }^{⊤}w_{1}\\ \Phi_{\delta }^{⊤}w_{2}\\ \vdots \\ \Phi_{\delta }^{⊤}w_{D_{w}} \end{array}\right]}}+N\left(0,\underset{R_{t}}{\underset{︸}{\left[\begin{array}{cccc} \sigma_{1}^{2} & 0 & \cdots & 0\\ 0 & \sigma_{2}^{2} & \cdots & 0\\ \vdots & \vdots & \ddots & 0\\ 0 & 0 & 0 & \sigma_{D_{w}}^{2} \end{array}\right]}}\right),$$
and the partial derivative of $h(\mu_{t})$ with respect to the weight vector $x_{t}={\left[w_{1},w_{2}\cdots ,w_{D_{w}}\right]}^{⊤}$ is: (18)$$H_{t}=\frac{\partial h(\mu_{t})}{\partial x_{t}}=\left[\begin{array}{ccc} \frac{\partial \Phi_{\delta }^{⊤}w_{1}}{\partial w_{1}} & \ldots & \frac{\partial \Phi_{\delta }^{⊤}w_{1}}{\partial w_{D_{w}}}\\ \vdots & \ddots & \vdots \\ \frac{\partial \Phi_{\delta }^{⊤}w_{D_{w}}}{\partial w_{1}} & \ldots & \frac{\partial \Phi_{\delta }^{⊤}w_{D_{w}}}{\partial w_{D_{w}}} \end{array}\right]=\left[\begin{array}{ccc} \Phi_{\delta }^{⊤} & \cdots & 0\\ \vdots & \ddots & \vdots \\ 0 & \cdots & \Phi_{\delta }^{⊤} \end{array}\right].$$

Finally, by combining the measurement model and the state transition model with the Bayesian filtering methodology, the expansion parameters of the LMO are updated continuously by observing the trajectories of human/object.

## 5. Dual Space and Time Simultaneous Inference with EnKF

The previous two sections mainly focused on generating robot motions based on human motion trajectories; that is, modeling and inferring the spatial uncertainty of robot motion. Spatial uncertainty estimation is the process of inferring the motion state of the robot at each moment. By using the spatial uncertainty inference in the previous sections, we can obtain the robot’s motion trajectory at any interaction time step. However, the robot does not know which point on the trajectory it should move to at this time. Therefore, we need to model and infer the temporal uncertainty of robot motion, which is essentially modeling and estimating the phase factor of the robot’s current stage; that is, phase estimation [32].

We draw inspiration from the BIP method, which extends the Kalman Filter to perform spatiotemporal synchronization inference, and we propose an inference method that can infer the phase, the linear mapping operator, and the dual-space trajectory weights simultaneously.

### 5.1. Spacetime Joint Modeling

In line with the modeling idea of this paper, the phase estimation problem can be formulated as how to find or estimate the value of $\xi \left(t\right)=\Phi {\left(t\right)}^{⊤}w$ at the current time step *t*. We assume that the phase of the demonstration is a constant-speed change process, and for a sample of length *T*, its phase velocity $\dot{\delta }=1/T$, and let β be the average phase velocity obtained from the demonstrations.

We define the complete state matrix as follows, by concatenating the phase and phase velocity: (19)$$s_{t}={\left[\delta_{t},{\dot{\delta }}_{t},\ell_{t},w^{⊤}\right]}^{⊤},s_{t}\in R^{D_{w}B+omn+2},$$
where $\delta_{t}$ and ${\dot{\delta }}_{t}$ are the phase and phase velocity at time *t*, respectively; $\ell_{t}$ is the parameter of the LMO; $w^{⊤}$ is the weight of the dual-space trajectories point. We initialize the phase estimation as follows: (20)$$\mu_{0}={\left[0,\beta ,\mu_{\ell },{\mu_{w}}^{⊤}\right]}^{⊤}.$$

To integrate phase estimation into the probabilistic inference of spatial trajectory, Equation (Equation 18) can be rewritten as: (21)$$\begin{array}{cc} H_{t} & =\left[\begin{array}{ccccc} \frac{\partial \Phi_{\delta }^{⊤}w_{1}}{\partial \delta } & \frac{\partial \Phi_{\delta }^{⊤}w_{1}}{\partial \dot{\delta }} & \frac{\partial \Phi_{\delta }^{⊤}w_{1}}{\partial w_{1}} & \ldots & \frac{\partial \Phi_{\delta }^{⊤}w_{1}}{\partial w_{D_{w}}}\\ \vdots & \vdots & \vdots & \ddots & \vdots \\ \frac{\partial \Phi_{\delta }^{⊤}w_{D_{w}}}{\partial \delta } & \frac{\partial \Phi_{\delta }^{⊤}w_{D_{w}}}{\partial \dot{\delta }} & \frac{\partial \Phi_{\delta }^{⊤}w_{D_{w}}}{\partial w_{1}} & \ldots & \frac{\partial \Phi_{\delta }^{⊤}w_{D_{w}}}{\partial w_{D_{w}}} \end{array}\right]\\ & =\left[\begin{array}{ccccc} \frac{\partial \Phi_{\delta }^{⊤}w_{1}}{\partial \delta } & 0 & \Phi_{\delta }^{⊤} & \cdots & 0\\ \vdots & \vdots & \vdots & \ddots & \vdots \\ \frac{\partial \Phi_{\delta }^{⊤}w_{D_{w}}}{\partial \delta } & 0 & 0 & \cdots & \Phi_{\delta }^{⊤} \end{array}\right] \end{array}.$$

The above process couples the phase estimation and the recursive state estimation of the dual-space trajectory weights. However, most of the existing studies use DTW to achieve phase estimation, which has a time complexity of $O(N^{2})$ [14]; DTW needs to query and compare the entire observed trajectory sequence at each time step, so it requires extra computational overhead for phase estimation while performing trajectory inference. We combine phase estimation with state recursion, which can avoid this extra overhead; for the overall algorithm, the two phase factors ($\delta_{t},{\dot{\delta }}_{t}$) introduced actually only increase two dimensions on the state matrix (Equation (Equation 19)), which has negligible impact on the overall state estimation dimension.

### 5.2. Spacetime Synchronization Inference

The linear mapping operator and the dual-space weights increase the dimension of the state matrix significantly. If we use the EKF inference method, we need to maintain a large covariance matrix at all times, and this high-dimensional matrix operation will bring more computational overhead. Therefore, we use the ensemble Kalman Filter (EnKF) inference method to handle the high-dimensional problem. EnKF does not need to maintain the covariance matrix like EKF, but uses an ensemble to simulate the state uncertainty.

The core idea of EnKF is as follows: we use an ensemble of dimension *E* to approximate the state prediction with the Monte Carlo method, where the sample mean of the ensemble is μ, and the sample covariance matrix is Σ. Each element of the ensemble is obtained by the state transition model, and is perturbed by stochastic process noise: (22)$$x_{t|t-1}^{j}=Gx_{t-1|t-1}^{j}+N\left(0,Q_{t}\right),\;1\leq j\leq E,$$
where $x_{t|t-1}$ denotes the prior of the state at time *t* obtained from time t−1. When *E* approaches infinity, the ensemble will effectively simulate the covariance calculation result of Equation (Equation 16). In the scenarios of pHRI, each element in the ensemble comes from the demonstration in the training dataset. The state ensemble is transformed to the measurement space through the nonlinear observation equation, and the error between each element and the sample mean in the ensemble is calculated: (23)$$H_{t}X_{t|t-1}=\left[h\left(x_{t|t-1}^{1}\right),\ldots ,h\left(x_{t|t-1}^{E}\right)\right],$$
(24)$$H_{t}A_{t}=H_{t}X_{t|t-1}-\left[\frac{1}{E}\sum_{j=1}^{E}h\left(x_{t|t-1}^{j}\right),\ldots ,\frac{1}{E}\sum_{j=1}^{E}h\left(x_{t|t-1}^{j}\right)\right].$$

Then, the new covariance matrix is: (25)$$S_{t}=\frac{1}{E-1}\left(H_{t}A_{t}\right){\left(H_{t}A_{t}\right)}^{⊤}+R_{t}.$$

The Kalman gain is calculated as follows: (26)$$A_{t}=X_{t|t-1}-\frac{1}{E}\sum_{j=1}^{E}x_{t|t-1}^{j},$$
(27)$$K_{t}=\frac{1}{E-1}A_{t}{\left(H_{t}A_{t}\right)}^{⊤}S_{t}^{-1}.$$

The ensemble is updated and the measurement results with random noise are considered: (28)$$Y_{t}=\left[y_{t}+\epsilon_{y}^{1},\ldots ,y_{t}+\epsilon_{y}^{E}\right],$$
(29)$$X_{t|t}=X_{t|t-1}+K_{t}\left(Y_{t}-H_{t}X_{t|t-1}\right).$$
when $\epsilon_{y}∼N\left(0,R_{t}\right)$, the ensemble can accurately represent the error covariance of the optimal state estimation [37]. The measurement noise is calculated by the following: (30)$$R_{t}=\frac{1}{N}\sum_{i}^{N}\frac{1}{T_{i}}\sum_{t}^{T_{i}}{\left(y_{t}-h\left(\left[\delta_{t},w_{i}\right]\right)\right)}^{2}.$$

Equations (Equation 22)–(Equation 29) describe the complete filtering process. In the scenarios of pHRI, we combine them with the EnKF mentioned above to implement the simultaneous generalization and probabilistic fusion of robot motion trajectories in dual space. The pseudocode of integrating dual-space trajectories, LMO, and phase estimation is presented in Algorithm 1.

**Algorithm 1:** Dual-space fusion algorithm for real-time pHRI task.

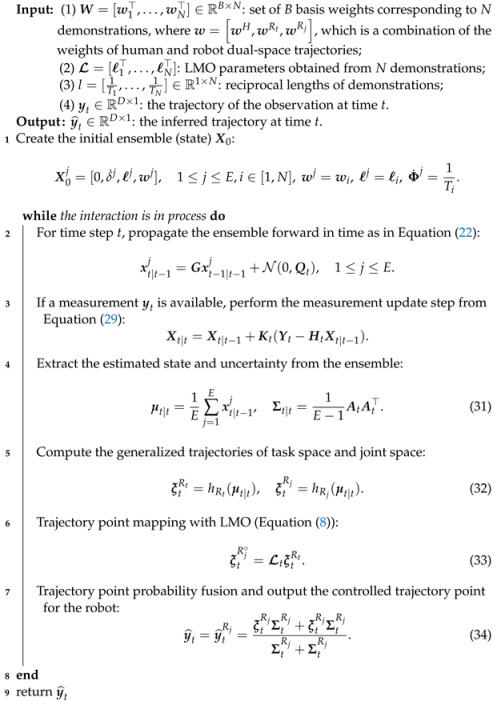



## 6. Experiments

To evaluate the proposed method, we conducted five experiments that cover different aspects of the problem:(1)Handwriting experiment in Section 6.1.1, where we simulated the dual-space features-learning capability, and mapping capability of LMO between different spaces by separating the X, Y, and Z axes of coordinates and treating them as three distinct spaces.(2)Human-robot-following experiment in Section 6.1.2, where we tested the ability of the probabilistic fusion method to learn both task space and joint space features. We also compared the trajectory error of different methods quantitatively.(3)Performance of LMO with different orders in Section 6.2, where we quantified the effect of the Taylor expansion order on the linearization performance of LMO.(4)Performance of phase estimation with different filters in Section 6.3, where we evaluated the error and computation time of phase estimation using different filtering techniques.(5)Experiment with higher real-time requirements, where we verified that the proposed method can meet the real-time constraints of the task and respond to the fusion of dual-space features.

### 6.1. Learning of Dual-Space Feature

In this part, we will demonstrate the capability of the learning task space feature and joint space feature of the proposed method, two experiments are included in this subsection: (1) a handwriting experiment and (2) a human-robot-following experiment.

#### 6.1.1. Handwriting Experiment

The illustration of this experiment is shown in Figure 2. We generated a handwriting dataset with 100 trajectories based on an opensource one [30], which are visualized in Figure 3a and their distributions are shown in Figure 3b. (The original dataset is two-dimensional (2D). We introduced a random variation in the Z-axis to create a three-dimensional (3D) version of the dataset.) Because this is a simulation experiment, the actual kinesthetic teaching (the leftmost of the top row in Figure 1) and the actual observation process (the leftmost of the bottom row in Figure 1) are no longer needed in this experiment. Instead, the trajectories provided by the dataset are used directly. We abstract an observed agent and a controlled agent, where the X-axis trajectories represent the observed agent, and the Y-axis and Z-axis trajectories represent the controlled agent. That is, the Y-axis and Z-axis trajectories can be regarded as two spatial features of the controlled agent. We think that there is a strong correlation between the different dimensions, and their relationship is constructed with our probabilistic fusion model mentioned in Section 3.

Then, the trajectories of the controlled agent (Y- and Z-axis) are inferred according to the observation (X-axis). We also observe that the two spaces (Y- and Z-axis) of the controlled agent are clearly correlated, even though they have no kinematic relationship. So that we use Equation (Equation 8) to estimate the parameters of LMO that capture the correlation between these two different spaces, the illustration of this experiment is shown in Figure 2.

We set the number of basis function components in each dimension to 9. The inference results are shown in Figure 4. We can see that a 3D data trajectory is generated based on the training dataset, which has not been observed before. Furthermore, the trained model infers the remaining motion trajectories of the controlled agents in two spaces (Y and Z) based on the observed partial motion trajectories of the observed agent. This validates that the proposed method can learn and infer different space features and high-dimensional data trajectories that do not have kinematic relationships.

#### 6.1.2. human-Robot-following Experiment

We also performed a real human-robot-following task which is shown in Figure 5. We collected a total of 60 demonstrations, where 45 were used for training and the remaining 15 for testing. The trajectories were formatted and divided into three parts: the trajectories of the human-held object (observed object), the end-effector trajectories of the UR5 manipulator (task space trajectories), and the trajectories of each joint angle of the UR5 manipulator (joint space trajectories). The three types of trajectories are modeled by fitting Gaussian basis functions of different numbers and scales (the scale is defined as the width of the basis [7]). In interactive inference, our algorithm performs real-time observation of the target object, simultaneously inferring the trajectories in both spaces and generating fused trajectories to control the robot to satisfy both task space and joint space requirements.

The top row of Figure 6 presents the process of observing human movement and inferring collision-free trajectories in both joint space and task space with our method. It can be found that the robot can perform the following task with human guidance and its joints go well around the box, i.e., the robot can satisfy the common constraints of both spaces and the dual-space trajectories obtained by synchronous inference are feasible and valid.

As a comparison, we use the same training set (excluding robot joint space trajectories) and Gaussian basis functions to model only task space inference in the same way as the IP framework, the parameters are shown in Table 2. In this case, the IP is only inferred for the task space control trajectory of the robot, its results are shown in the bottom row of Figure 6. It can be seen that the robot follows the target object when only task space inference is performed. For the sake of controlling variables, when performing the experiment of IP, we replay the observed trajectories of the object recorded at the experiment of our method; that is, the data played back here are the same as the trajectory data of the human-held object in the top row of Figure 6. It can be seen that considering only the task space requirements (IP), the robot is not guaranteed to fulfill the joint space constraints (the robot knocks over the box). On the contrary, motion generated with our method can not only accurately achieves the goal, but also learns the joint space features effectively and ensures the collision-free requirements. This verifies the effectiveness of the proposed method in learning dual-space features.

We also quantified the error of the inferred trajectories using different methods and spaces. We first modeled the inference of the joint space only with IP. Then, we compared the performance of three inference approaches: task space only (IP), joint space only (IP), and dual-space fusion (ours). We used the mean absolute error (MAE) and root mean squared error (RMSE) metrics to evaluate the trajectories generated by each approach on the same test set. Table 3 shows the results. For each group of demonstrations in the test set, we observed the target object trajectories (human actions) and generated the robot control trajectories using each of the three approaches in both spaces. We then computed the RMSE and MAE between the robot trajectories of the test set (demo trajectories) and the trajectories generated by each approach in each space at each time step.

Let *M* be the number of demonstrations (that is, M=15) in the test set. Assume that there are $N_{i}$ interaction classes. (An interaction class of a time step is defined as a tuple of the observed object’s position, end-effector’s position, and robot’s joint angles.) In the *i*-th ($i\in \left[1,M\right]$) demonstration, let $p_{j}^{A}$ be robot end-effector’s coordinates in the *j*-th interaction class of *i*-th demonstration; that is, $p_{j}^{A}=\left[x_{j}^{A},y_{j}^{A},z_{j}^{A}\right]$ (A for actual). Furthermore, let $p_{j}^{I}$ be the corresponding coordinates that are predicted with different methods; that is, $p_{j}^{I}=\left[x_{j}^{I},y_{j}^{I},z_{j}^{I}\right]$ (I for inferred). Then, the MAE and RMSE for the task space are defined as: (35)$$MAE=\frac{1}{M}\sum_{i=1}^{M}\frac{1}{N_{i}}\sum_{j=1}^{N_{i}}\left(|x_{j}^{I}-x_{j}^{A}\left|+\right|y_{j}^{I}-y_{j}^{A}\left|+\right|z_{j}^{I}-z_{j}^{A}|\right),$$
(36)$$RMSE=\frac{1}{M}\sum_{i=1}^{M}\sqrt{\frac{1}{N_{i}}\sum_{j=1}^{N_{i}}\left(∥p_{j}^{I}-p_{j}^{A}{∥}_{2}^{2}\right)}.$$

For the joint space, let $\theta_{j}^{A}$ be the robot’s joint angles in the *j*-th interaction class of *i*-th demonstration; that is, $\theta_{j}^{A}=\left[\theta_{j1}^{A},\theta_{j2}^{A},\ldots ,\theta_{jK}^{A}\right]$, where *K* is the degrees of freedom of the robot, as well as the inferred joint angles $\theta_{j}^{I}=\left[\theta_{j1}^{I},\theta_{j2}^{I},\ldots ,\theta_{jK}^{I}\right]$. Then, the MAE and RMSE for the joint space are defined as: (37)$$MAE=\frac{1}{M}\sum_{i=1}^{M}\frac{1}{N_{i}}\sum_{j=1}^{N_{i}}\frac{1}{K}\sum_{k=1}^{K}\left|\theta_{jk}^{I}-\theta_{jk}^{A}\right|,$$
(38)$$RMSE=\frac{1}{M}\sum_{i=1}^{M}\sqrt{\frac{1}{N_{i}}\sum_{j=1}^{N_{i}}\left(∥\theta_{j}^{I}-\theta_{j}^{A}{∥}_{2}^{2}\right)}.$$

As shown in Table 3, the proposed method achieves the best performance in all comparison metrics. The joint space trajectories have large deviations (*RMSE*: 3.4 × 10${}^{-1}$ rad) when only the task space trajectories are inferred, which is consistent with the result shown in Figure 6. When only the joint space is inferred, the robot can satisfy the joint space constraint (*RMSE*: 8.5× 10${}^{-2}$ m), but its end-effector trajectory accuracy (*MAE*: 2.2 × 10${}^{-2}$ m) is worse than that of our dual-space fusion method (*MAE*: 8.4 × 10${}^{-3}$ m). Moreover, UR5 is a 6 DoF manipulator, so its end-effector position accuracy is less influenced by joint generalization than that of a redundant robot.

As demonstrated by the above experiments, our dual-space probabilistic fusion approach can handle HRI tasks with joint constraints in both task space and joint space and enhance the accuracy of single-space inference in either task space or joint space.

### 6.2. Performance of LMO with Different Orders

It is impossible to expand Equation (Equation 10) to an infinite series because the more that the series are expanded, the more parameters are involved, and the higher the computational time consumption. Therefore, we conducted this experiment to investigate the effect of the order of LMO on inference accuracy, and finally determine how many series should be expanded.

The same dataset as the human-robot-following task in Section 6.1.2 is used here. We compared our method with different orders (first to fifth order) of LMO to our previous work (baseline) [13], in which the robot kinematics are used. In our method, we approximate the kinematics mapping from the task space to the joint space by LMO. To ensure a fair comparison, the same number of basis function components and the same scaling factor in basis functions are applied for all methods. Furthermore, the EnKF is replaced with EKF which is the same as the baseline. The MAE and the RMSE are calculated to measure the error between the inferred trajectories in robot joint space and the ground truth, and the quantitative results are summarized in in Figure 7. The results show that the MAE decreased by 33% and the RMSE decreased by 37% when LMO is switched from the first order to the second order. However, increasing the order of LMO further does not lead to significant reductions in MAE and RMSE. This is because the Taylor expansion is a method of approximating a function by an infinite series. The more terms in the series, the closer the approximation is to the true value of the function. However, as the number of terms increases, the convergence rate of the approximation slows down. The MAE of second-order LMO is 20% higher than that of the baseline, and the RMSE is 15% higher than that of the baseline. From the above relative comparison, it can be seen that the inference accuracy of LMO is somewhat lower than that of the baseline (using the robot’s kinematics model directly). However, from the absolute values, the second-order LMO is very close to the baseline, and both MAE and RMSE maintain a very small value, which can meet the requirements of most HRI tasks.

### 6.3. Performance of Phase Estimation with Different Filters

This experiment consists of two parts: the first part is to compare and analyze the inference accuracy and phase estimation accuracy of different methods (IP, EKF, and EnKF), and the second part is to compare the inference computation with using different filters (EKF and EnKF). In each group of comparison experiments, training data, samples, and parameter settings are the same.

Figure 8 compares and evaluates the phase estimation error and trajectory accuracy of EKF, EnKF, and the original IP. The gray line in the figure shows the mean phase error of the entire interaction process obtained by EKF for the testing set, and the gray area shows the ensemble range formed by different testing samples. Similarly, the red line and red area show the mean phase error and ensemble range obtained by EnKF for the same testing set. The right vertical axis in the figure indicates the phase estimation error obtained by different filters. Comparing EKF, EnKF, and the original IP, we obtain three mean error lines at the bottom of Figure 8. These three error lines and Table 4 reveal that the dual-space feature inference of the Kalman Filter framework is more accurate than the original IP, which is consistent with the conclusion of human-robot-following in Section 6.1.2. Moreover, EnKF achieves better results than EKF, but their results are overall similar, which is also reflected in their phase estimation comparison. This is because they share a similar modeling idea for phase estimation, so in the interaction process, for the same training set, their inference mean lines and distribution situations are very close. Due to the introduction of phase parameters and LMO, the dimension of the state vector has increased significantly. If we use EKF, it involves a large-scale operation process on the covariance matrix, and its computational complexity is $O(N^{3})$, where *N* denotes the dimension of the state vector. For EnKF (this paper), we use a set of dimension *E* to represent the state matrix, which is essentially a kind of dimension reduction, and its computational complexity is reduced to $O(E^{2}N)$. The selected set size (*E*) is generally much smaller than the state dimension (*N*), which will contribute to improving the performance of computation.

The dataset recorded in the last experiment is input into the algorithms in the form of data playback, simulating an interaction scenario, and then recording the time required for each iterative inference. The computational cost is obtained, visualized, and analyzed in Figure 9 and Table 5.

As shown in Figure 9, the iteration speed of EnKF is generally higher than that of EKF, and it can also be found that the time required for EKF is unstable and fluctuates greatly (the gray line in the figure has obvious sawtooth fluctuations). On the contrary, EnKF has a relatively stable inference time at each interaction moment, and its overall variance is small. As can be seen from Table 5, compared with EKF, the overall inference speed of EnKF is increased by 54.87%. Combined with the results of the previous experiment, it can be concluded that the EnKF spatiotemporal synchronous inference method adopted in this paper achieves higher improvement in computational performance while ensuring inference accuracy, and is more suitable for real-time inference in HRI.

### 6.4. An Experiment with Higher Real-Time Requirements

In order to demonstrate the real-time inference ability of the proposed framework for HRI tasks, we conducted a human-robot collaborative ball-hitting experiment, as shown in Figure 1 and Figure 10. Two human participants were involved in the data collection process. One participant threw a ball from different distances and angles, whereas the other participant demonstrated the hitting motion to a KUKA iiwa7 robot with a racket attached to its end-effector via kinesthetic teaching. Trajectories of the ball (task space) were recorded by an Optitrack motion capture system at 120 Hz. The robot’s joint motion trajectories (joint space) and its end-effector pose trajectories (task space) were recorded via ROS. Furthermore, a total of 140 training samples were collected. The fitting parameters of each type of trajectory were determined by BIC, and the specific Gaussian radial basis function parameters are shown in Table 6.

The snapshots of the live human-robot ball-hitting experiment are shown in Figure 11. In the live interaction, the trained model simultaneously infers the robot motion trajectories in the robot joint space and task space according to the observed ball motion trajectories and controls the robot to complete the hitting. If the inference accuracy and real-time performance are insufficient, it will cause the robot to miss the hitting point. The four pictures in Figure 11a together show that our inference algorithm observes the ball’s trajectory and generates the response trajectory for the robot. Furthermore, the robot synchronizes its movement with the ball’s parabolic motion and hits the ball (task space) at the predicted point using a similar configuration (joint space) as the demonstrations. Similarly, Figure 11b shows a higher landing point for the ball trajectory, and the robot makes a downward hitting action with a racket. The two interaction processes demonstrate our online adaptive ability and verify that our inference performance can perform real-time pHRI tasks. The video of the human-robot-following experiment and ball-hitting experiment is Appendix A.

## 7. Conclusions

This paper proposes a probabilistic dual-space (task space and joint space) feature fusion for real-time physical human-robot interaction. We conducted several qualitative and quantitative experiments and analyses to validate our effectiveness. Notably, we achieved:A pHRI method that considers the dual-space features of the robot. The dual-space trajectory inference fusion method proposed in this paper addresses the limitation of the original IP, which can only perform single-space generalization. By applying our method with two datasets of a UR5 and a KUKA iiwa7 robot, we reduce the trajectory inference error of the proposed method by more than 33% in both spaces. In addition, we demonstrate the applicability of the proposed method to different robots by verifying it with two datasets of robots with different DoFs. Moreover, the inferred results satisfy both task space and joint space constraints and can meet real-time inference interaction requirements.A dual-space linear mapping method that does not depend on the robot model. We propose a method to learn and infer the linear mapping relationship between different spaces from demonstrations. Through the three-dimensional handwriting trajectory experiment, we verify the feasibility of integrating the linear mapping operator into the dual-space synchronous inference framework. With the UR5 dataset experiment, we examine the influence of the expansion order of the linear mapping operator on the trajectory inference accuracy, and we show that increasing the order of the mapping operator within a certain range can improve the inference trajectory accuracy. In the experiment, the trajectory MAE obtained by the second-order LMO is close to that obtained by the kinematic mapping.We integrate phase estimation into the Kalman Filter framework to realize a method for spatiotemporal synchronous inference, which reduces the inference hierarchy. Based on the idea of ensemble sampling, we use ensemble Kalman Filter inference to solve the high-dimensional inference problem caused by dual-space inference. The experiments show that our improvements maintain the accuracy superiority of dual-space feature learning, and that the overall computational efficiency is 54.87% higher than EKF.

Our method learns both the task space and joint space features of the demonstration, enabling it to handle scenarios that require both target position and configuration, especially for real-time tasks. As humanoid robots and related technologies become more integrated into people’s daily lives, our method can be applied to service robots to generate motions that satisfy simultaneous constraints on multiple robot joint spaces and task spaces, such as robot anthropomorphic writing, dancing, sign language imitation and communication, greeting, etc. These tasks not only depend on the position of the robot’s hand (end-effector), but also on whether the robot’s configuration (shape) is human-like. In the future, we will explore using our method to learn more feature fusion, such as speed, acceleration, force, etc., so that the generated motion exhibits more human-like characteristics, enhancing its applicability in practical HRI and improving the interaction efficiency and human acceptance of robots.

## Figures and Tables

**Figure 1 biomimetics-08-00497-f001:**
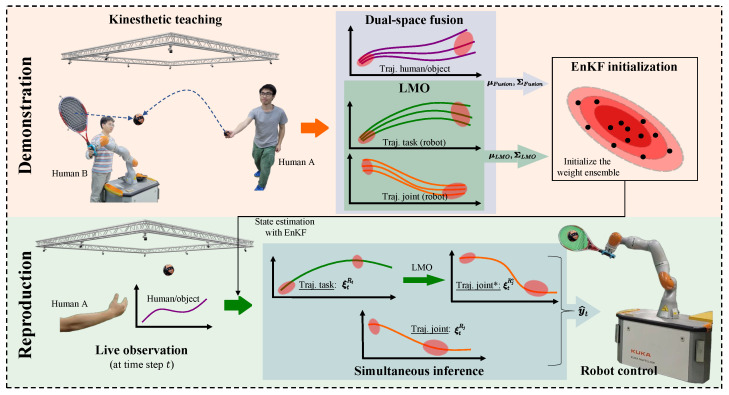
Overview of the proposed method. **Top**: During the demonstrations, human A performs some actions (such as throwing a ball in the figure), human B teaches the robot to respond to Human A’s actions through kinesthetic teaching, and the motion capture system synchronously records the task space and joint space trajectories of the robot, as well as the task space trajectories of Human A or object (for example, in the hitting experiment (Section 6.4), the task space trajectory of the ball thrown by Human A is recorded, and in the object following experiment (Section 6.1.2), the task space trajectory of the human-held object is recorded). At the **dual-space fusion** (Section 3) block, the demonstrations (all the trajectories) are decomposed with several Gaussian basis functions and obtain parameters of the weight distribution; at the **LMO** (Section 4) block, the mapping from task space to joint space is linearized by Taylor expansion, and then the parameters of its series are solved by Gaussian distribution, obtaining the parameters of the distribution. All the distribution parameters are transformed into a fusion ensemble of samples (right). **Bottom**: during live interaction, the robot generates a response trajectory (right) to the observation of human A/object with the learned fusion ensemble and LMO (Section 5).

**Figure 2 biomimetics-08-00497-f002:**
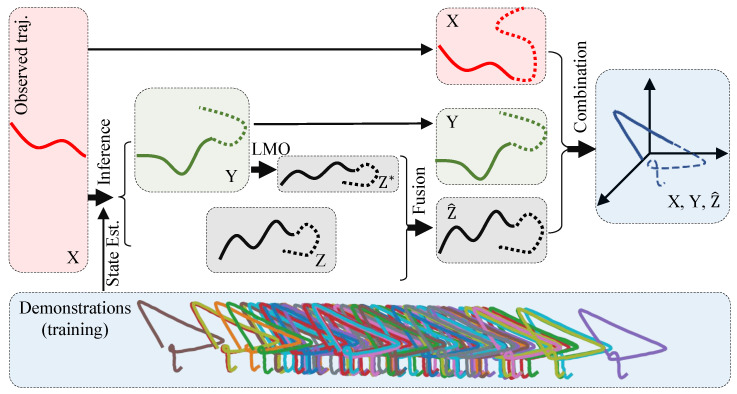
Illustration of the handwriting experiment. We consider the X-, Y-, and Z-axis of the trajectories’ coordinates as three distinct spaces, corresponding to different aspects of the human-robot interaction. The X-space represents the task space trajectory of the human or object, the Y-space represents the task space trajectory of the robot, and the Z-space represents the joint space trajectory of the robot, as shown in Figure 1. We train our model with trajectories from all three spaces as input. After training, we can infer a new Y-space trajectory and a new Z-space trajectory by observing a partial X-space trajectory. We can also map the new Y-space trajectory to another Z-space trajectory (Z*) using LMO. The final Z-space trajectory ($\hat{Z}$) is obtained by fusing Z and Z*. Finally, we combine the X, Y, and $\hat{Z}$ trajectories into a 3D trajectory and compare it with the demonstration.

**Figure 3 biomimetics-08-00497-f003:**
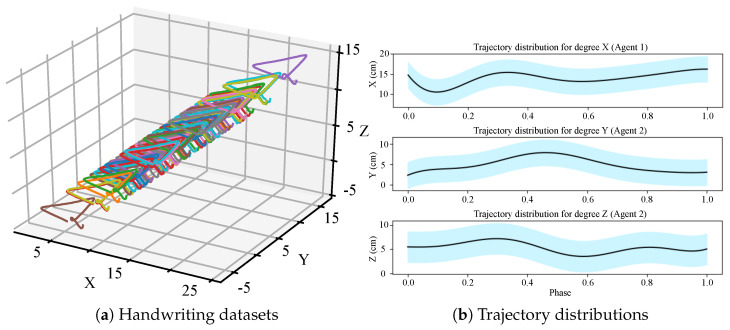
The handwriting trajectories and inference. (**a**) The 100 3D trajectories of handwriting, that is those curves with different colors in the figure; (**b**) the trajectory distributions of X (**top**), Y (**middle**), and Z (**bottom**) axes.

**Figure 4 biomimetics-08-00497-f004:**
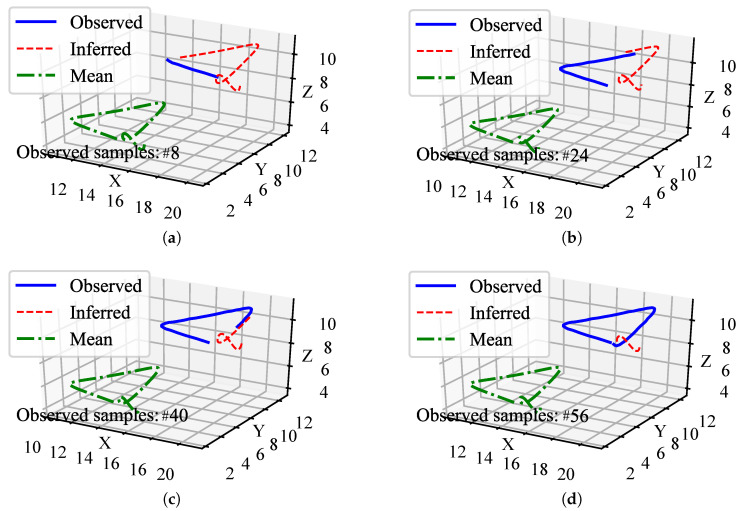
Examples of the handwriting trajectories inference. Four examples (**a**–**d**) are visualized with different observed samples (#8, #24, #40, and #56). The green dashed lines depict the mean of data trajectories. The blue solid lines depict the observed data trajectories for testing. Furthermore, the red dotted lines depict the inferred data trajectories. Since we only observed the trajectories in the X-axis, coordinates in one direction cannot be visualized in a three-dimensional coordinate system, the observed trajectories shown in the figures are combined with both the X-, Y-, and Z-axis of the observed sample, but the trajectories in Y- and Z-axis are not observed in the whole inference process.

**Figure 5 biomimetics-08-00497-f005:**
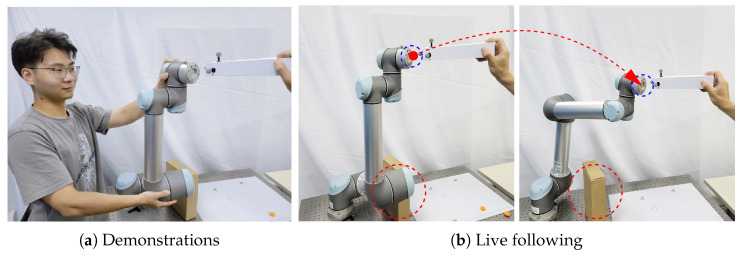
Illustration of the human-robot-following task. (**a**): The robot was demonstrated to follow the target object and avoid colliding with the obstacle. (**b**): during the live interaction, a new guidance was performed by the user (the red arrow), and the robot can generated a collision-free (the red circle) trajectory with the proposed method.

**Figure 6 biomimetics-08-00497-f006:**
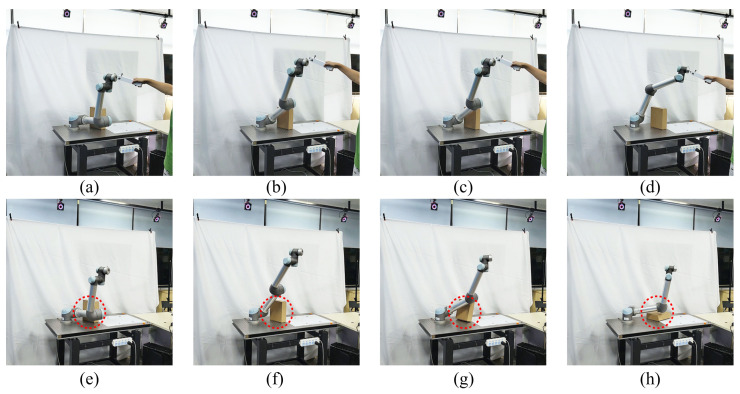
Snapshots of the human-robot-following experiment. **Top**: the interaction process of the proposed method. (**a**–**d**) represent screenshots of different moments of the process from start to stop, and (**e**–**h**) do the same. The end-effector of the UR5 manipulator follows the movement of a human handheld object while fulfilling the joint space requirements. **Bottom**: the interaction process that is derived from using IP to model and infer the trajectory of the task space. The red circle depicts that there is a collision between the robot and the obstacle. To ensure that both methods use the same observed trajectory for inference, we set the observed task trajectory to be identical to ours. This means that, in the live interaction of IP, we replay the trajectory of the observed object that was recorded in the experiment using our proposed method.

**Figure 7 biomimetics-08-00497-f007:**
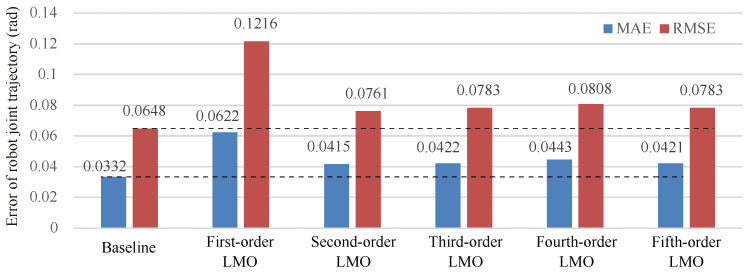
Comparisons of inference accuracy with different orders of LMO. The baseline is our previous work [13] in which the robot kinematics are used.

**Figure 8 biomimetics-08-00497-f008:**
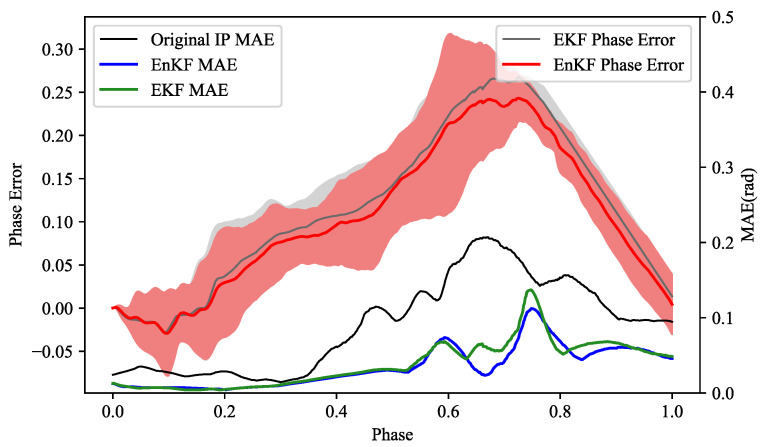
Comparison of the inferred performance of different methods.

**Figure 9 biomimetics-08-00497-f009:**
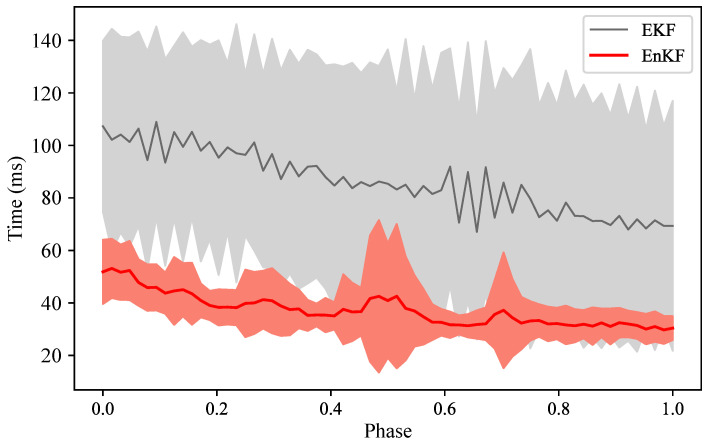
Comparison of the computational performance between the two methods.

**Figure 10 biomimetics-08-00497-f010:**
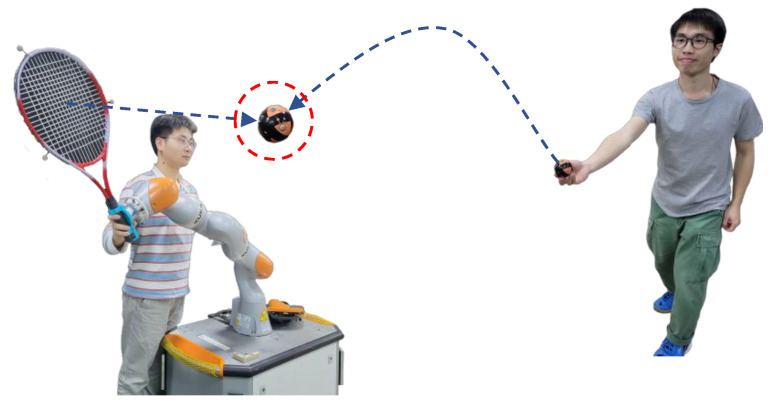
Illustration for the human-robot ball-hitting experiment. The blue dotted line depicts the motion trajectories of the ball and the racket, and the red circle depicts hitting point. Several reflective markers are attached to the ball and the racket to capture their motion.

**Figure 11 biomimetics-08-00497-f011:**
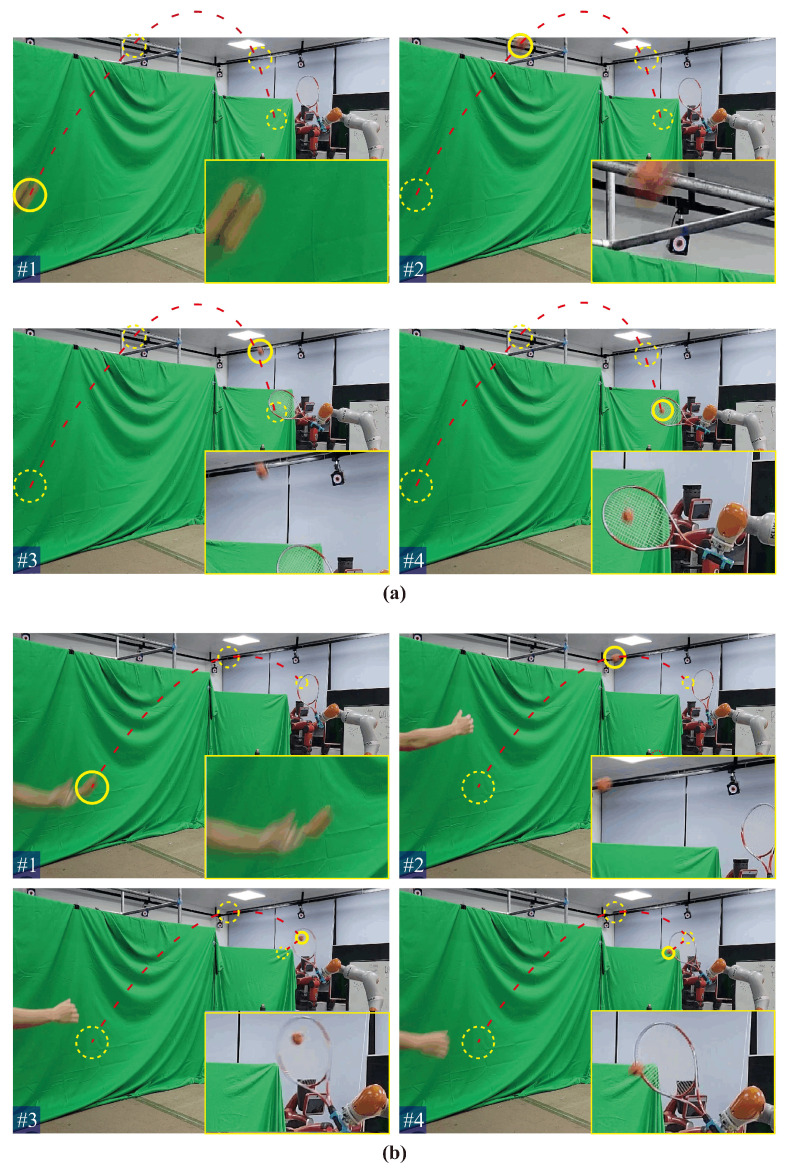
Snapshots of the live human-robot ball-hitting task. We present two groups (**a**,**b**) of hitting experiments in the figure, each containing four sub-figures (#1, #2, #3, and #4). The red dotted lines represent the ball motion trajectories, the yellow solid circles represent the current position of the ball, and the yellow dashed circles represent the position of the ball at other moments. In addition, a partially enlarged view is placed at the right bottom corner of each sub-figure, which shows the current state of the ball, e.g., the one in #4 is the scene of the ball being hit by the racket.

**Table 1 biomimetics-08-00497-t001:** Related works comparison in required terms.

Method or Author	HRI	Spatial Uncertainty Modeling	Time Uncertainty Modeling	Dual-Space Fusion	Meet Real-Time Interaction Requirements
GMM/GMR [5]	**✗**	**✓**	**✗**	**✗**	**✗**
HMM/HSMM [4]	**✗**	**✓**	**✗**	**✗**	**✗**
DMP [6]	**✗**	**✗**	**✗**	**✗**	**✗**
ProMP [7]	**✗**	**✓**	**✗**	**✗**	**✗**
GP [34]	**✗**	**✗**	-	**✗**	**✗**
KMP [8]	**✓**	**✓**	-	**✗**	**✗**
IP [14]	**✓**	**✓**	**✗**	**✗**	**✓**
BIP [30]	**✓**	**✓**	**✓**	**✗**	**✓**
TMG [25]	**✗**	**✓**	**✗**	**✗**	**✗**
Calinon et al. [22]	**✗**	**✓**	**✗**	**✗**	**✗**
Schneider et al. [26]	**✗**	**✓**	**✗**	**✗**	**✗**
Huang et al. [29]	**✗**	**✓**	**✗**	**✓**	**✗**
Requirement	**✓**	**✓**	**✓**	**✓**	**✓**

**✓**: yes, **✗**: no, -: unnecessary.

**Table 2 biomimetics-08-00497-t002:** Basis function parameters of trajectories for the human-robot-following task.

Trajectory	Number of Basis Functions	Scale of Basis
Task space trajectory (object)	13	0.01
Task space trajectory (robot)	9	0.01
Joint motion trajectory (robot)	9	0.04

**Table 3 biomimetics-08-00497-t003:** Comparison of trajectory errors among three inference methods.

	Task Space		Joint Space	
	**RMSE (m)**	**MAE (m)**	**RMSE (rad)**	**MAE (rad)**
Task space only (IP)	4.3 × 10${}^{-2}$	2.9 × 10${}^{-2}$	3.4 × 10${}^{-1}$	9.9 × 10${}^{-2}$
Joint space only (IP)	3.5 × 10${}^{-2}$	2.2 × 10${}^{-2}$	1.3 × 10${}^{-1}$	6.9 × 10${}^{-2}$
Dual-space fusion (ours)	**1.2 × 10${}^{-2}$**	**8.4 × 10${}^{-3}$**	**8.5 × 10${}^{-2}$**	**4.0 × 10${}^{-2}$**

**Table 4 biomimetics-08-00497-t004:** Comparison of the inference accuracy among the three methods.

Metrics (rad)	Original IP	EKF	EnKF
MAE	9.08 × 10${}^{-2}$	3.82 × 10${}^{-2}$	**3.44 × 10${}^{-2}$**

**Table 5 biomimetics-08-00497-t005:** Comparison of the mean computational costs between EKF and EnKF.

Metrics	EKF	EnKF
mean cost time (ms/epoch)	83.17	**37.39**

**Table 6 biomimetics-08-00497-t006:** Basis function parameters of modeling for ball-hitting experiment.

Trajectory	Number of Basis Functions	Scale of Basis
Ball position trajectory	13	0.01
Robot pose trajectory	9	0.01
Robot joint motion trajectory	9	0.04

## Data Availability

All data related to this paper will be made available upon request. The access to the data is subject to the data protection policy of the Biomimetic and Intelligent Robotics Lab (BIRL) and Guangdong University of Technology (GDUT), China.

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
