# Peer review of "Probabilistic Dual-Space Fusion for Real-Time Human-Robot Interaction"

_biomimetics, 2023, doi:10.3390/biomimetics8060497_

Round 1

Reviewer 1 Report

On Conclusions, discussing the Future research work, bring more value to this research study. 

The quality of English Language is average.

Reviewer 2 Report

In the presented work, the authors proposed a method of dual-space feature fusion to enhance the accuracy of inferred trajectories tacking task space and joint space into account. The authors then introduced a linear mapping operator (LMO) to map the inferred task space trajectory to a joint space trajectory and combined the dual-space fusion, LMO, and phase estimation into a unified probabilistic framework. They evaluated the dual-space feature fusion capability and real-time performance in the task of a robot following a human-handheld object and a ball-hitting experiment. The inference accuracy in both task space and joint space was superior to standard Interaction Primitives (IP) by more than 33%, which only use single-space inference of the second order LMO and the computation time of the unified inference framework reduced by 54.87%.

 The proposed method seems to be useful. The authors should justify the relationship between the proposed visual model in Figure 1 and the corresponding mathematical models in Sections 3, 4 and 5. It means the authors should explain how the mathematical models in Sections 3-5 properly and adequately address the visual model in Figure 1. Similarly, the authors should justify how the experimental results (algorithmic results) truly and adequately justify the concepts proposed in the mathematical models in Sections 3-5. It is not clear what the authors wanted to mean by pHRI. The authors should identify the most relevant pHRI criteria considering the literature and then assess them properly to justify the effectiveness of their mathematical models for pHRI. What about cognitive HRI?

It is necessary that the authors show a 3-dimensional tradeoff among the accuracy, stability and efficiency of the proposed dual-space feature fusion method. Currently, it is missing.

At least one photo of the handwriting experiment is required to understand the experimental procedures. In general, each experiment should be presented with all procedural steps sequentially.

The authors should present the future directions of their research.

Minor editing of English language required.

Reviewer 3 Report

The paper describes a imitation learning method that is using two redundant target trajectory spaces which then are combined by means of Bayesian inference, while the redundant part of the trajectory space is projected into the target space by means of an estimated linearization of the transfer function.

There is one question that has not been answered in the whole paper.

Why is a distinction of task space and joint space necessary at all? Would the task space target (which I understood is the end effector position) not directly inherit from a joint space target?

The benefit of the combined method over joint space only imitation learning also with your second experiment is not proofed.

I put a couple of remarks and questions in the PDF directly. Please find them atached.

The English language is ok, but there are a couple of minor typos to be fixed.

Round 2

Reviewer 2 Report

The authors tried their best to address the points raised by the reviewer. The reviewer appreciates the efforts made by the authors. The revision certainly improved the overall quality of the manuscript. The authors wrote their responses to the points raised by the reviewer seriously.  However, the authors did not revise/improve the manuscript adequately based on their responses to review comments.

Particularly, the authors' justification of the relationship between the proposed visual model in Figure 1 and the corresponding mathematical models in Sections 3, 4 and 5 is still unclear and inadequate. It means the authors need to further explain how the mathematical models in Sections 3-5 properly and adequately address the visual model in Figure 1 and revise the manuscript accordingly. Similarly, the authors need to further justify how the experimental results (algorithmic results) truly and adequately justify the concepts proposed in the mathematical models in Sections 3-5.

The authors explained what they wanted to mean by pHRI. However, the authors did not use the most relevant pHRI and cHRI criteria to assess the models properly to justify the effectiveness of their mathematical models.

In the revised version, the authors tried to show a 3-dimensional tradeoff among the accuracy, stability and efficiency of the proposed dual-space feature fusion method. The cross-relationship/tradeoff among the accuracy, stability and efficiency of the proposed dual-space feature fusion method needs to be explained properly.

Figure 2 is an illustration (diagram) for the handwritten experiment. It is not a photo showing the actual experimental method/environment/scenario.

Minor editing of English language required.
